# Effect of Terpenoids and Flavonoids Isolated from *Baccharis conferta* Kunth on TPA-Induced Ear Edema in Mice

**DOI:** 10.3390/molecules25061379

**Published:** 2020-03-18

**Authors:** Gutiérrez-Román Ana Silvia, Trejo-Tapia Gabriela, Herrera-Ruiz Maribel, Monterrosas-Brisson Nayeli, Trejo-Espino José Luis, Zamilpa Alejandro, González-Cortazar Manasés

**Affiliations:** 1Centro de Desarrollo de Productos Bióticos. Instituto Politécnico Nacional (IPN), Col. San Isidro, Carretera Yautepec-Jojutla, Km 6, 62731, Morelos, Mexico; 12gtr.ana@gmail.com (G.-R.A.S.); jtrejo@ipn.mx (T.-E.J.L.); 2Centro de Investigación Biomédica del Sur, Instituto Mexicano del Seguro Social, Argentina 1, Col. Centro, Xochitepec, 62790 Morelos, Mexico; cibis_herj@yahoo.com.mx (H.-R.M.); azamilpa_2000@yahoo.com.mx (Z.A.); 3Facultad de Ciencias Biológicas, Universidad Autónoma del Estado de Morelos (UAEM), Av. Universidad 1001, Col. Chamilpa, Cuernavaca, 62209 Morelos, Mexico; nmonterrosas78@gmail.com

**Keywords:** diterpenes, flavonoids, topical anti-inflammatory, bacchofertone, *Baccharis conferta* Kunth

## Abstract

In this study, we isolated from the aerial parts of *Baccharis conferta* Kunth (i) a new neoclerodane, denominated “bacchofertone”; (ii) four known terpenes: schensianol A, bacchofertin, kingidiol and oleanolic acid; and (iii) two flavonoids: cirsimaritin and hispidulin. All structures were identified by an exhaustive analysis of nuclear magnetic resonance (NMR) and mass spectroscopy (MS). Extracts from aerial parts were screened for anti-inflammatory activity in the mice ear edema model of 12-*O*-tetradecanoylforbol-13-acetate mice. Dichloromethane extract (BcD) exhibited 78.5 ± 0.72% inhibition of edema, followed by the BcD2 and BcD3 fractions of 71.4% and 82.9% respectively, at a dose of 1 mg/ear. Kingidiol and cirsimaritin were the most potent compounds identified, with a median effective dose of 0.12 and 0.16 mg/ear, respectively. A histological analysis showed that the topical application of TPA promoted intense cell infiltration, and this inflammatory parameter was reduced with the topical application of isolated compounds.

## 1. Introduction

*Baccharis* L. (Asteraceae) includes approximately 400 species that are distributed from USA to Argentina. This genus is very relevant in folk medicine because many of these species have been used medicinally (e.g., to reduce phlegm, relieve cough, improve blood circulation, reduce pain, induce diuresis, as anthelmintic, anti-inflammatory and stomachic agents). In recent years, 27 species of the genus *Baccharis* have been chemically investigated, and 139 compounds corresponding to flavonoids, triterpenes, cinnamic acids and diterpenoids have been isolated. Neoclerodanes diterpenes are characteristic constituents from this genus and possess a great anti-inflammatory effect in acute and chronic inflammatory conditions. Twenty-four diterpenoids have been reported, and 16 new terpenoids have been isolated, including seven neoclerodanes [1,2,3,4,5,6,7]. Hautriwaic acid, a neoclerodane diterpene, showed anti-inflammatory activity on two murine models: ear edema induced with 12-*O*-tretradecanoylphorbol-13-acetate (TPA) and monoarthritis induced with kaolin/carrageenan [8,9]. *Baccharis conferta* Kunth (Asteraceae), commonly called “escoba” or “escobilla china”, is used in Mexican traditional medicine for the treatment of join pain, seizures, cramps, toothache, colds and digestive disorders [10,11]. Previous reports have indicated that an ethanol extract of *B. conferta* containing flavonoids has an antispasmodic effect. On the other hand, a methanol extract showed ovicidal activity against *Haemonchus contortus*, where the active compounds were isokaemferide and 4,5-di-*O*-caffeoylquinic. Phytochemical investigations of *B. conferta* have reported the isolation of essential oils, diterpene, triterpenes, flavonoids and coumarins [12,13,14,15].

The aim of this study was to determine the anti-inflammatory activity from *B. conferta* and to identify the compounds responsible for this effect through bioassay-guided fractionation, using the TPA-induced ear edema model in mice.

## 2. Results and Discussion

### 2.1. Anti-Inflammatory Effect of Extracts and Fractions of B. conferta

Mouse ear edema induced by TPA has been used to test anti-inflammatory activity. The TPA, when applied to the skin, causes a rapid and potent irritant effect by activating the protein kinase C (PKC) pathway, which results in elevated levels of eicosanoids such as prostaglandin E_2_ (PGE_2_) and leukotriene B_4_ (LTB_4_). In addition, it promotes the activation of the MAPKs pathway, which induces the release of pro-inflammatory cytokines such as interleukin (IL)-1, tumor necrosis factor (TNF)-α, IL-8 and macrophage inflammatory protein (MIP)-1β. Adhesion molecules and enzymes are activated, resulting in the formation of edema and the migration of leukocytes into the dermis [16]. The anti-inflammatory activities of the BcH, BcD and BcM extracts of *B. conferta* using the TPA-induced ear edema model in mice are shown in Table 1.

The administration of 1 mg/ear of BcD extract, which is mainly composed by moderately polar compounds, had the highest percentage of inhibition, i.e., 78.5%. Organic extracts and fractions of several species of the *Baccharis* genus have been investigated pharmacologically for their anti-inflammatory properties using different in vitro models. With different degrees of activity, all extracts from *B. obtusifolia*, *B. latifolia*, *B. pentlandii*, *B. subulata* inhibited PGE2-release (COX-2), NO-release and TNF-α-release, among other inflammatory mediators; these data indicated the potential anti-inflammatory effects of species from this genus [17].

To find the active principles responsible for this anti-inflammatory activity, the BcD active extract was separated into three main fractions, i.e., BcD1, BcD2 and BcD3, which were tested in a TPA-mouse ear edema assay. The BcD1 fraction was the least polar and contained mainly fatty acids. The BcD2 and BcD3 fractions decreased edema by 4.54 ± 0.52 mg and 2.53 ± 0.75 mg, representing 71.4% and 82.9% inhibition of inflammation respectively, and were not significantly different from the effect of treatment with indomethacin (Table 1). The BcD2 and BcD3 fractions were less complex and more active than the BcD extract. Thus, they were fractionated by column chromatography, which led to the isolation and characterization of seven compounds (**1**–**7**).

### 2.2. Isolation and Elucidation of Compounds 1–7

This bioassay-guided chemical investigation led to the identification of seven bioactive compounds (Figure 1).

Compound **1** was obtained as a colorless oil with UV spectra (MEOH) λ_max_ (log ε) 207.7 nm. In the positive-ion ESI-MS of 1, ion peaks at m/z 277 [M + Na]^+^ (calcd. for C_15_H_26_O_3_Na, 277.21), indicating the molecular formula C_15_H_26_O_3_. According to these NMR data analyses (See Appendix A) and their comparison with data described previously, this compound was identified as (3S, 7R, 10S)-3, 11-dihydroxy-7, 10-epoxy-3, 7, 11-trimethyldodeca-1, 5-diene, commonly known as schensianol A (**1**). This compound has been previously isolated from *Euonymus schensianus* [18] and *Echinacea purpurea*; like *Baccharis*, *E. purpurea* belongs to the Asteraceae family [19]. This is the first time that schensianol A and its anti-inflammatory effect have been reported from *B. conferta*.

Compound **2** was obtained as colorless crystals with a melting point of 169–170 °C. It showed one absorption band at λ_max_ 218 nm and its molecular formula was determined based on the ESI-MS data (m/z 331 [M + H]^+^ calcd., C_20_H_27_O_4_, 331.2). The ^13^C NMR spectrum of **2** showed 20 characteristic signals for a diterpene, and the ^1^H NMR spectrum (Table 2 and Table 3) displayed the following representative signals: three signals of double bonds and a doublet at δ 6.28 with *J* = 1.9 Hz that is coupled in COSY with another signal at δ 7.37 (dd, *J* = 1.2, 1.9 Hz); this, in turn, had a wide singlet signal at δ 7.24 that was assigned to the protons H-14, H-15 and H-16 of the β-substituted furan ring. Their correlations in HSQC correspond to δ 110.79, 142.94 and 138.42, respectively (Figure 2a).

The presence of two methyl groups was also observed: a singlet signal at δ 0.89 that correlates in carbon with the signal at δ 18.58 assigned to C-20 and another doublet signal at δ 0.86 (*J* = 6.4 Hz) which is assigned to C-17 (δ 15.28). The presence of α, β-unsaturated lactone is evident in this compound from the signals at δ 6.57 (dd, *J* = 1.9, 6.4 Hz) of the olefinic β-protons in C-3, which correlate in HSQC with signal at δ 129.22 and with carbonyl at long-range at δ 169.8 (C-18) and, in turn, in HMBC with the signals at δ 4.63 (dd, *J* = 2.56-7.69 Hz) and δ 4.33 (d, *J* = 7.69 Hz) for oxymethylenes at C-19 (δ 73.3) (Figure 2b). The H-3 (δ 6.57) correlates in the COSY experiment with two proton signals of a methylene at δ 2.49 (ddd, 3.2, 5.7, 18.5) and 2.44 (dd, 3.8, 18.5) assigned to H-2, which is coupled with a broad singlet signal of oxygen base at δ 4.44 assigned to H-1. The axial α-orientation of the hydroxyl group was corroborated by the NOESY interaction of the H-1 signal (δ 4.44, s) with H-10 (1.41, d, 1.9 Hz) (see Figure 2b).

The analysis of the spectroscopic data allowed us to identify this compound as a diterpene of the neoclerodane type called (6S,6aR,7S,8R,10aS)-7-[2-(furan-3-yl)ethyl]-6-hydroxy-7,8-dimethyl-5,6,6a,8,9,10-hexahydro-1H-benzo[d][2]benzofuran-3-one (**2**), known as bacchofertin, identified by Guerrero and Romo de Vivar (1973) in *B. conferta*. No information has been published about any biological effect of this compound.

Compound **3** was obtained as a colorless oil with one absorption band at λ_max_ 218 nm; its molecular formula was determined based on the ESI-MS data (m/z 341 [M + Na]^+^ calcd., C_20_H_30_O_3_Na, 341.26). Analysis of the NMR spectra showed that this compound is similar to compound **2**, having the same signals, except for the hydrolysis of lactone of C-18 its reduction (C=O) to alcohol (CH_2_OH) by the presence of the signals at δ 4.21 (d, *J* =11.3 Hz) and 3.84 (d, *J* = 11.3 Hz) assigned to H-18a and H-18b, as well as the absence of the hydroxyl in C-1 to a methylene that was observed at δ 4.21 (d, *J* = 11.3 Hz) and 3.84 (d, *J* = 11.3 Hz. All NMR spectroscopic data were obtained by one- (^1^H and ^13^C) and two-dimensional (COSY, HSQC and HMBC) experiments (Table 2 and Table 3), which allowed us to determine that this compound corresponds to the neoclerodane known as kingidiol (**3**), identified only in *Baccharis kingii*, which has been evaluated as a juvenile hormone antagonist [20,21].

Compound **4** was obtained as a white powder with one absorption band at λ_max_ 240 nm; its molecular formula was determined based on the ESI-MS data (m/z 457 [M + H]^+^ calcd., C_30_H_49_O_3_, 457.36). According to these NMR data analyses and their comparison with data described previously [22], this compound was identified as oleanolic acid (**4**). In the *Baccharis* genus, this terpene has been described in *B. linearis* [23], *B. illinita* [24], and *B. conferta* [2]. The oleanolic acid isolated was evaluated in the TPA-induced ear edema model by Boller et al. (2010) in *B. illinita* [25].

Compound **5** was obtained as colorless crystals with one absorption band at λ_max_ 218 nm; its molecular formula was determined based on the ESI-MS data (m/z 347 [M + H]^+^ calcd., C_20_H_27_O_5_^+^, 347.24). An analysis of the NMR spectra (Table 2 and Table 3) showed that this compound is similar to compound **2**, with the difference being that compound **5** presents another α-β-unsaturated lactone, which was corroborated by the IR absorption in 1730 cm^−1^. The signals of ^1^H-NMR at δ 7.1 (s, br) for oleofinic β-proton at C-14 (δ 143.9) correlate in COSY with the signal at δ 4.8 (s, br), which is assigned to an oxymethylene at C-15, and in HMBC, correlated with a carbonyl at δ 174.0 (Figure 3). According to the analysis of the spectroscopic data, this compound corresponds to 1-hydroxy-neo-clerodan-3, 13-dien-16-dien-16, 15: 18, 19-diolide, which is a derivate of mkapwanin isolated from *Dodonaea angustifolia* [26], a new diterpene which we denominate as bacchofertone (**5**).

Compound **6** was isolated as a yellow powder with a melting point of 260–262 °C. In the UV light spectrum, the compound showed λmax at 213, 254, 273 and 344 nm, which is characteristic of a flavone. In the positive-ion ESI-MS of 6, ion peaks at m/z 315 [M + H]^+^ (calcd. for C_17_H_15_O_6_, 315.14). A direct comparison of the spectroscopic data (see Appendix A) with the literature allowed us to identify this compound as 5-hydroxy-2-(4-hydroxyphenyl)-6, 7-dimethoxychromen-4-one also called cirsimaritin (**6**) [27], isolated from *B. conferta* by Weimann et al., 2002 [12].

Compound **7** was isolated as a yellow powder with a melting point of 260–262 °C. In the UV light spectrum, the compound showed λmax at 213.4, 254, 273 and 344 nm, characteristic of a flavone. In the positive-ion ESI-MS of 7, ion peaks at m/z 301 [M + H]^+^ (calcd. for C_16_H_13_O_6_, 301.12). An analysis of the NMR spectra showed that this compound is similar to compound **6** (see Appendix A), having the same signals, except for the absence of a C-7 methoxy. This compound was identified as 5, 7-dihydroxy-2- (4-hydroxyphenyl) -6-methoxychromen-4-one or called hispidulin (**7**) [28]. Hispidulin has previously been isolated from plant species of the *Baccharis* genus, especially from *B. pseudoternuifolia*, *B. uncinella*, *B. flabellate*, *B. trimera* and *B. gaudichaudina* [2], and its anti-inflammatory activity has been investigated [29,30].

### 2.3. Anti-Inflammatory Effect of Compounds of B. Conferta

Pharmacological evaluation of all compounds except **4** (because the anti-inflammatory effect thereof has already been reported [31]) in a TPA-induced ear edema model in mice showed anti-inflammatory action, with a more than 70% reduction in edema at a dose of 1 mg/ear. Compounds 3 and 6 showed the highest percentage of inhibition of edema, with 94.1 ± 0.9% and 98.1 ± 0.06%, respectively (Table 1). Because the compounds showed significant anti-inflammatory effects, dose-response curves were performed (Figure 4). In this concentration-dependent behavior, the pharmacological constants were calculated as E_max_ and ED_50_ (Table 4). According to these data, compounds **3** and **6** are the most potent, having a low DE_50_, while **2** is the least potent. Compounds **6** and **7** are known to display anti-inflammatory activity in croton oil-induced ear edema, which is regulated by the inhibition of interleukin-6, tumor necrosis factor-α, NO production [32], c-fos and STAT3 phosphorylation in a concentration-dependent manner in lipopolysaccharide (LPS)-stimulated RAW264.7 cells [33].

### 2.4. Histological Analysis of Ear Edema-Induced by TPA

Figure 5 shows a microscopic representation of the effect of TPA on cell architecture, as well as the activity of the different compounds isolated from *B. conferta*. Six hours after TPA application, the ear observations indicated an intense dermal edema and increase in inflammatory cells. This includes, for example, high infiltration of neutrophils within the epidermis present primarily in the intermediate and higher levels and epidermal thickening, though there were some areas of focal thinning of the epidermis. Within the upper, intermediate, and deep dermis, neutrophils were scattered. There was marked dermal edema with dilated vascular lymphatic spaces and a mild perivascular mononuclear infiltrate of lymphocytes and monocytes compared with animals whose epidermises were normal. The irritation induced by the topical administration of TPA produced the rupture of capillaries, and the infiltration of erythrocytes below the epidermis and ears that received acetone did not show any damage or edema.

Compound **3** at 1 mg/ear presented the best anti-inflammatory activity in the TPA assay (see Table 1); however, an ear histological analysis showed more inflammatory cell infiltration and lymphatic spaces than the other treatments evaluated here (Figure 5). This may be because compound **3** was more effective in reducing edema than in protecting tissue from the damage induced by TPA. Even so, the tissue damage was less than that of the VEH group.

Compound **6** showed a significant inhibition effect on TPA-induced ear edema (see Table 1). Histological analysis showed only a few inflammatory cells within the epidermis, but a slight perivascular neutrophilic inflammatory infiltrate was present with much less edema. The tissue returns to a normal structure comparable with the basal group (Figure 5).

The compounds with the least anti-inflammatory effect in the TPA-induced edema assay (see Table 1) were **1**, **2**, **5** and **7**. The ears from mice of these treatments showed neutrophils within the epidermis primarily in the intermediate and higher levels, and there were some areas with slight focal epidermal thickening. Compound 5, unlike the others, showed lymphatic spaces similar to those of the VEH group. With compounds 1 and 5, the erythrocyte infiltrates were present, similar to the VEH group. However, the tissue was less damaged compared to the VEH group (Figure 5)

## 3. Materials and Methods

### 3.1. General Experimental Procedures

Melting points were obtained using a Thermo Scientific IA9000 series melting point apparatus and were uncorrected. Compounds (1–6) NMR spectra were recorded on an Agilent DD2-600 at 600 MHz and compound 7 in a Bruker Advance III HD-600 at 600 MHz for 1H NMR, NOESY, ^1^H-^1^H COSY, HSQC and HMBC, and 150 MHz for ^13^C NMR and DEPT in CDCl_3_ and CD_3_OD. Chemical shifts are reported in ppm relative to TMS. Mass spectroscopy analysis was performed on a Waters Xevo TQD mass spectrometer with an ESI ion source (Waters Milford, USA). The ultraviolet (UV) spectra were obtained using a Waters array detector (Waters Co. 2996, Milford, USA). Thin-layerchromatography (TLC) was performed using TLC Silica gel 60, F_254_, and 20 × 20 cm aluminium sheets (Merck KGaA, Darmstadt, Germany). High-performance liquid chromatography (HPLC) analysis was performed on a Waters 2695 Separation module system, equipped with a Waters 996 photodiode array detector and Empower Pro software (Waters Corporation, USA). The ultra-performance liquid chromatography coupled with a mass detector (UPLC-MS) analysis was performed on an Acquity UPLC (Waters, Milford MA, USA). HPLC-PDA analysis was achieved using a Supelcosil LC-F column (4.6 mm × 250 mm i.d., 5 µm particle size) (Sigma-Aldrich, Bellefonte, USA). The mobile phase consisted of 0.5% trifluoroacetic acid aqueous solution (solvent A) and acetonitrile (solvent B). The gradient system was as follows: 0–1 min, 0% B; 2–3 min, 5% B, 4–20 min, 30% B; 21–23 min, 50% B; 24–25 min, 80% B; 26–27 100% B; 28–30 min, 0% B. The flow rate was maintained at 0.9 mL min^−1^ with a sample injection volume of 10 µL and wavelength range detection of 190–600 nm. Diterpene compounds were analyzed at 220 nm and flavonoids at 350 nm. UPLC-MS analysis was performance an Acquity (Waters, Milford MA, USA) system. This separation system included a quaternary pump, auto-sampler column oven and a photodiode array-detector coupled with a “Xevo” (Waters) triple quadrupole mass spectrometer equipped with an electrospray ionisation (ESI) source (Waters) heated at 150 °C. The desolvation temperature was set at 500 °C and the desolvation gas flow was 700 L/h nitrogen. Argon was used as a collision gas at a flow rate of 0.10 mL/min. (Thermo Fisher Scientific, Bremen, Germany). Chromatographic separations were performed with an Acquity UPLC BEH 1.7 m-C18 column at a flow rate of 0.3 mL/min. The column was eluted with 0.1% trifluoroacetic acid aqueous solution (A) and 0.1% trifluoroacetic in acetonitrile (B). The column was held at 100% of A for 1 min and subsequently ramped to 100% of B over 11 min, followed by a 4 min period at 100% of B before a rapid return to 100% of A, and an equilibration period of 2 min. The column was maintained at a temperature of 40 °C and injection volume was 3 µL.

### 3.2. Plant Material

Aerial parts of *B. conferta* (20 kg) were collected from the Iztaccíhuatl-Popocatépetl National Park (N 19 0.4’13.5 ‘‘, W 99 20’23.2 ‘‘, 3128 masl) in Mexico State, Mexico, in February 2018. The sample specimen of this material was identified by specialists at the National Autonomous University of Mexico (voucher code number: 150228); it was stored in the MEXU-UNAM Herbarium. The plant material was oven-dried at 40 °C for 3 days and pulverized in a Pulvex MPP300 mill to reduce the particle size to approximately 4–6 mm.

### 3.3. Preparation of Eextracts

Dry plant material (8 kg) was macerated with 10 L of n-hexane (Merck) in triplicate, filtered and vacuum-concentrated using a rotary evaporator (Heidolph G3) at 40 °C for later lyophilization (Heto Dpywinner DW3) until obtaining a powder called the hexanic extract (BcH). The dried plant residue was macerated with dichloromethane (10 L, Merk) and later with methanol (Merck) following the procedure described above to obtain the dichloromethane (BcD) and methanol (BcM) extracts. Three extracts were evaluated in the anti-inflammatory activity assay described below. The macerations resulted in the following amounts and yields for each extract: BcH (150.7 g, 1.80%); BcD (548.2 g, 6.85%) and BcM (750.9 g, 9.38%).

### 3.4. Chromatographic Fractionation of the Most Active Extract (BcD)

The most active extract BcD (300 g) was adsorbed in silica gel and applied to a silica gel gravity column (500 g, 70–230 mesh, Merck, Darmstadt, Germany). A gradient of n-hexane/ethyl acetate was used to elute the column with an increase in polarity of 10% per 2 L. This chromatographic process resulted in eight fractions: BcC1F1 (100:0, 5.2 g), BcC1F2 (90:10, 6.0 g), BcC1F3 (80:20, 7.2 g), BcC1F4 (70:30, 12.7 g), BcC1F5 (60:40, 17.6 g), BcC1F6 (50:50, 26.6 g), BcC1F7 (100:0, 112.8 g) and BcC1F8 (100% methanol, 142.3 g). The BcC1F3, BcC1F6 and BcC1F8 fractions were selected for evaluation in the anti-inflammatory model according to the chemical content presented, and are hereafter refered to as BcD1, BcD2 and BcD3, respectively.

### 3.5. Isolation and Identification of Compounds (1-4) from the BcD2 Active Fraction

The BcD2 fraction (2 g) was purified by successive open column reversed phase chromatography using RP-18 silica gel (10 g) as the stationary phase and a gradient of water/acetonitrile as the mobile phase. Forty-one fractions (15 mL) were obtained and grouped into twenty-two subfractions according to their chemical composition (BcC2F1-22). The BcC2F6 fraction (20 mg) was identified as schensianol A (1). The BcC2F14 fraction (500 mg) was submitted to chromatographic column, absorbed, placed in a silica gel column (10 g) and eluted with dichloromethane/methanol. Nineteen fractions were obtained (BcC3F1-BcC3F19). The BcC3F8 fraction (40 mg) was identified as bacchofertin (2). The BcC2F16 fraction (1 g) was submitted to a chromatographic column, absorbed, placed in an RP-18 silica gel column (7 g) and eluted with water/acetonitrile as the mobile phase. Six fractions were obtained (BcC4F1-BcC4F6). The BcC4F5 fraction (60 mg) was identified as kingidiol (**3**). The BcC2F22 fraction (100 mg) resulted in a white precipitate that was identified as oleanolic acid (**4**).

### 3.6. Isolation and Identification of Compounds (5-7) from the BcD3 Active Fraction

The BcD3 fraction (20 g) was purified by open column chromatography using silica gel as the stationary phase. A gradient of dichloromethane/methanol as the mobile phase was used to elute the column with an increase in polarity of 5% per 225 mL. This chromatographic process resulted in 17 fractions which were grouped according to the similarity of the compounds into six fractions (BcC5R1-BcC5R6). The BcC5R2 fraction (2 g) was a mixture of compounds; it was submitted to a chromatographic column, adsorbed in 0.5 g silica gel 60 and 0.5 g silica gel RP-18, placed in an RP-18 silica gel column (16 g) and eluted with water/acetonitrile as the mobile phase. This chromatographic process resulted in 30 fractions which were grouped according to their chromatographic similarity into eight fractions (BcC6R1-BcC6R8). The BcC6R2 fraction (20 mg) was identified as 1-hydroxyneoclerodane-3, 13-diene- 15, 16; 18, 19-diolide, which was called bacchofertone (5). The BcC6R3 (25 mg) and BcC6R8 (15 mg) fractions were identified as cirsimaritin (6) and hispidulin (7).

### 3.7. Experimental Animals

Male ICR mice (body weight range 25–30 g) were obtained from Envigo Mexico RMS (Mexico City, Mexico). All animals were housed under standard laboratory conditions (at a temperature of 22 ± 3 °C, with 70 ± 5% of humidity, with 12 h light/dark cycles and with food and water ad libitum). All experimental procedures were carried out in accordance with the guidelines established in the International and Official Mexican Standards for Animals Care and Health (NOM-062-ZOO-1999). Each mouse was used only once during the protocol and was euthanized in a chloroform chamber immediately after the experiments ended. For the experimental procedure, the groups (VEH or treatments) consisted of five animals each (n = 5).

Protocol registered with the Ethics Committee and research (R-2018-1702-013).

### 3.8. TPA-Induced Mouse Ear Edema

Mouse ear edema was induced following the method previously described by Payá et al. (2010) [34]. The treatments were assayed topically in the left ear of each mouse, and the right ear was used as a control. All the treatments were performed by applying 10 µL in the external and 10 µL in the internal ear. The treatments were divided into a negative control group, which received 2.5 µg/ear of TPA, and a positive control group, which received indomethacin (Sigma) at 1 mg/ear in acetone. Extracts, fractions and compounds (1–7, except 4) were evaluated at a dose of 1 mg/ear.

All treatments were dissolved in acetone and applied topically on both ears. Fifteen min later, the TPA solution was administered, and 4 h after the administration of the inflammatory agent, the animals were sacrificed by cervical dislocation. Circular sections 6 mm in diameter were taken from both the treated (t) and nontreated (nt) ears, which were weighed to determine the inflammation. Percentage of inhibition was obtained using the equation below:

Inhibition (%) = [Δw control−Δw treatment] × 100

where Δw = wt−wnt, wt is the weight of the section of the treated ear and wnt is the weight of the section of the non-treated ear.

A dose-response curve was performed for the anti-inflammatory compounds based on the following doses: 0.062, 0.125, 0.25, 0.5 and 1 mg/ear. We calculated the ED_50_ for each compound.

### 3.9. Histological Analysis

Ear sections were fixed in a 10% phosphate-buffered formaldehyde solution for one day, dehydrated in a series of ethanol solutions of increasing concentrations up to 100% and then cleared with xylene. The tissue was then embedded in paraffin, sectioned (5 μm) and stained with Hematoxylin and Eosin (H&E) procedure.

### 3.10. Statistical Analysis

The data are expressed as the means ± standard error of the mean (SEM), and statistical significance was determined using an analysis of variance (ANOVA) followed by Tukey‘s test comparing each treatment with the VEH and INDO groups. Values were considered significant at p ≤ 0.05.

## 4. Conclusions

In conclusion, our study shows that the dichloromethane extract (BcD) and fractions (BcD2 and BcD3) and compounds from the aerial parts of *B. conferta* have topical anti-inflammatory activity. The present work describes, for the first time, a new compound, denominated bacchofertone (5); schensianol A (1), kingidiol (3) and hispidulin (7) are known compounds described for other species but not for *B. conferta*. Compounds 1, 2, 3 and 5 demonstrated efficacy at reducing inflammatory parameters, such as edema and cell migration, in a TPA-induced ear edema model.

## Figures and Tables

**Figure 1 molecules-25-01379-f001:**
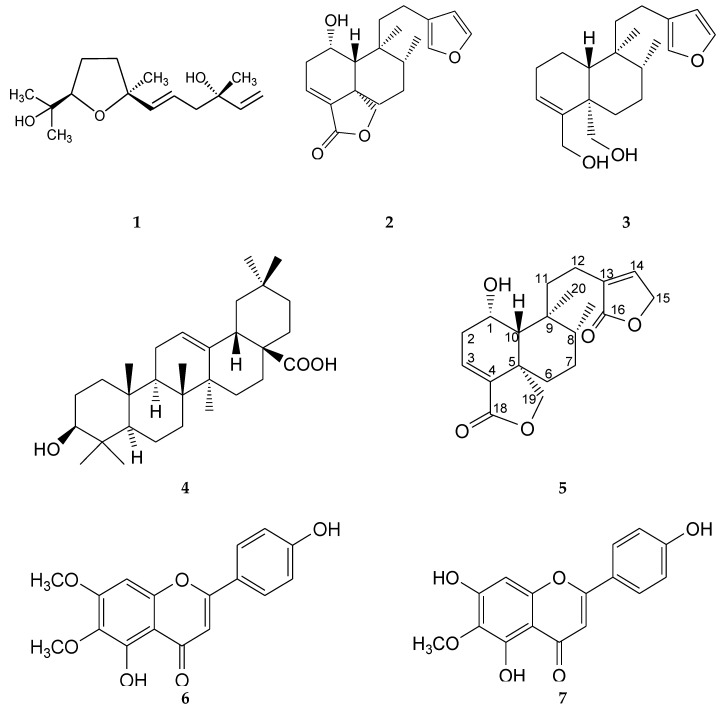
Chemical structure of compounds (**1**–**7**); schensianol A (**1**), bacchofertin (**2**), kingidiol (**3**), oleanolic acid (**4**), bacchofertone (**5**), cirsimaritin (**6**) and hispidulin (**7**) isolated of *B. conferta*.

**Figure 2 molecules-25-01379-f002:**
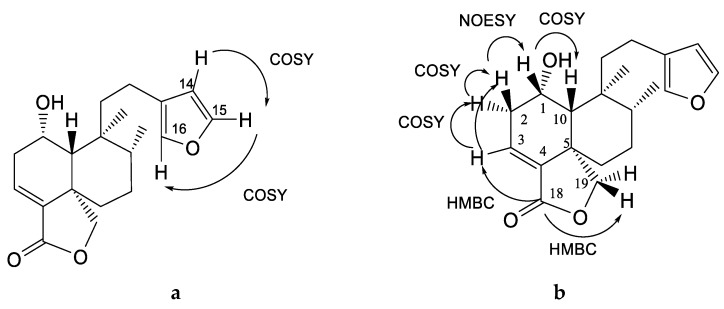
Correlations; (**a**) COSY and (**b**) COSY, NOESY and HMBC of compound **2**.

**Figure 3 molecules-25-01379-f003:**
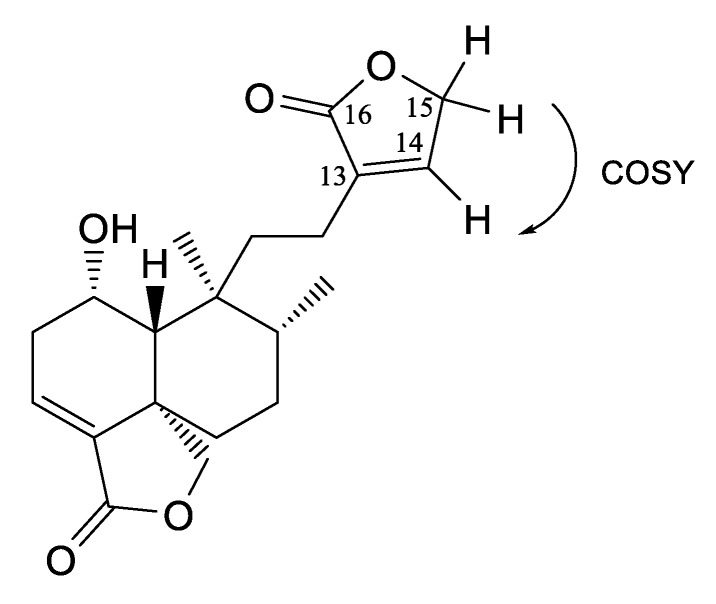
Correlations of compound **5**.

**Figure 4 molecules-25-01379-f004:**
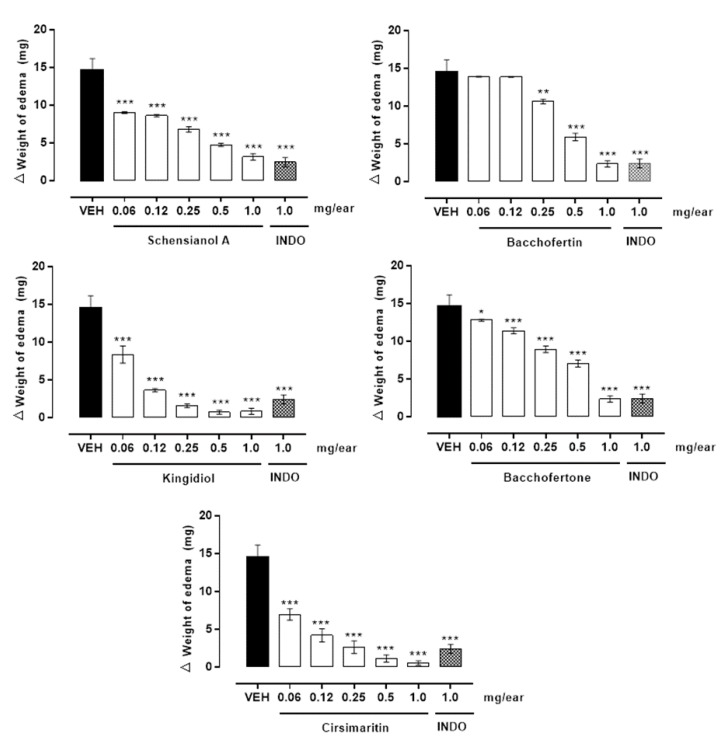
Curve dose-response of isolated compounds from *B. conferta* on TPA-induced ear edema. Each treatment with TPA (VEH) immediately before the application of schensianol A (**1**), bacchofertin (**2**), kingidiol (**3**), bacchofertone (**5**), hispidulin (**6**) and cirsimaritin (**7**). The bars show the mean ± S.E.M. of the weight difference in milligrams (mg) (n = 5) 6 h after TPA application. Significantly different from VEH, ^*^p ˂ 0.05, ^**^p ˂ 0.01, ^***^p ˂ 0.0001.

**Figure 5 molecules-25-01379-f005:**
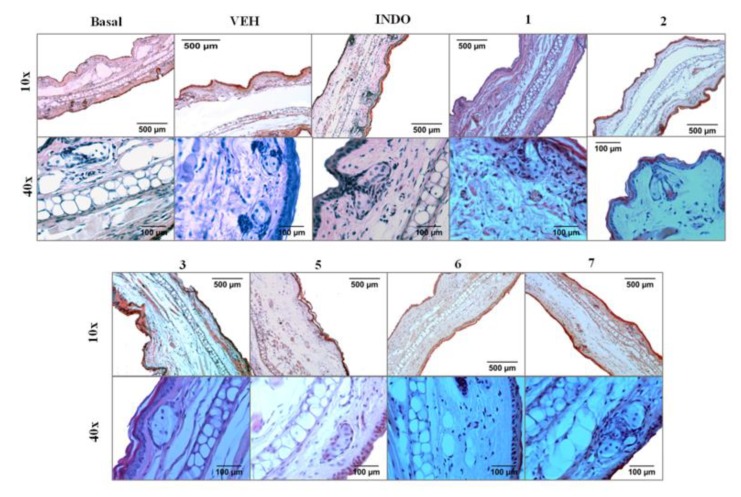
Representative pictures of histological transversal cuts of mice ears and analysis of alterations induced by TPA and effect different kinds of compounds isolated from *B. conferta* from mice ears stained with hematoxylin-eosin. Schensianol A (**1**), bacchofertin (**2**), kingidiol (**3**), bacchofertone (**5**), hispidulin (**6**) and cirsimaritin (**7**).

**Table 1 molecules-25-01379-t001:** Anti-inflammatory activity of extracts, fractions and compounds from *B. conferta.*

Substance	Edema (mg) mean ± SEM	Edema inhibition (%)
VEH	14.3 ± 1.5	---
BcH	9.5 ± 0.54 *^b^	42.83
BcD	3.08 ± 0.31 **	78.50
BcM	7.73 ± 0.11 *^c^	45.92
BcD1	9.78 ± 0.28 **^a^	31.64
BcD2	4.54 ± 0.23 **	68.29
BcD3	2.53 ± 0.33 **	82.28
**1**	3.12 ± 0.36 **	78.18
**2**	2.36 ± 0.45 **	83.50
**3**	0.84 ± 0.41 **	94.13
**5**	3.32 ± 0.39 **	76.78
**6**	0.27 ± 0.02 **	98.14
**7**	3.66 ± 0.5 **	74.41
INDO	1.22 ± 0.54 **	85.66

** p ˂ 0.0001, * p ˂ 0.001 in comparison with VEH group, ^a^p ˂ 0.0001, ^b^p ˂ 0.001, ^c^p ˂ 0.05 in comparison with INDO group.

**Table 2 molecules-25-01379-t002:** ^13^C-NMR spectroscopy data of **2**, **3** and **5** (CDCl_3_, 150 MHz).

Position	δC 2	δC 3	δC 5
**1**	64.3	17.4	63.2
**2**	37.2	27.4	36.7
**3**	129.2	129.4	129.9
**4**	138.0	145.1	137.4
**5**	44.9	43.0	44.7
**6**	35.5	31.3	35.3
**7**	27.7	26.9	27.5
**8**	37.4	36.5	37.2
**9**	39.3	38.9	39.0
**10**	49.8	46.4	49.8
**11**	37.6	38.7	35.0
**12**	18.3	18.3	18.9
**13**	124.8	125.5	133.9
**14**	110.7	110.9	143.9
**15**	142.9	142.7	70.3
**16**	138.4	138.4	174.0
**17**	15.2	15.9	14.9
**18**	169.8	64.3	169.6
**19**	73.2	65.6	73.3
**20**	18.5	18.8	18.0

**Table 3 molecules-25-01379-t003:** ^1^H-NMR spectroscopy data of **2**, **3** and **5** (CDCl_3_, 600 MHz).

Position	δ_H_ (*J* in Hz) 2	δ_H_ (*J* in Hz) 3	δ_H_ (*J* in Hz) 5
**1a** **1b**	4.43 (s, br)	1.64 (m) 1.61(m)	4.43 (d, br, 1.9)
**2a** **2b**	2.49(ddd, 3.2, 5.7, 18.5) 2.44 (dd, 3.8, 18.5)	2.31 (m) 2.17 (m)	2.46 (m) 2.46 (m)
**3**	6.57 (dd, 1.92, 6.41)	5.76 (dd, 3.6)	6.57 (dd, 2.6, 6.6)
**4**	-	-	-
**5**	-	-	-
**6a** **6b**	1.26 (ddd, 1.9, 3.8, 12.8) 1.9 (dd, 3.8, 12.8)	1.16 (dd, 7.3, 12.8) 2.27 (dd, 12.84, 17.6)	1.25 (m) 1.95 (dd, 1.9, 13.2)
**7a** **7b**	1.59 (dddd, 1.9, 3.8, 4.4, 12.8) 1.66 (m)	1.52 (dd, 5.5, 11.3) 1.46 (ddd, 2.9, 9.9, 11.7)	1.6 (m) 1.63 (m)
**8**	1.67 (m)	1.63 (m)	1.63 (m)
**9**	-	-	-
**10**	1.79 (s, br)	1.53 (m)	1.72 (s, br)
**11a** **11b**	1.82 (ddd, 4.4, 7.6, 12.1) 1.75 (dd, 5.1, 12.8)	1.52 (m) 1.63 (m)	1.85 (ddd, 3.9, 9.2, 13.2) 1.62 (m)
**12a** **12b**	2.43 (ddd, 5, 14, 17.9) 2.12 (ddd, 4.4, 13.4, 17.9)	2.29(m) 2.17 (m)	2.29 (ddd, 2.65, 4.64, 13.93) 1.98 (m)
**13**	-	-	-
**14**	6.28 (s, br)	6.25 (s, br)	7.1 (s, br)
**15a** **15b**	7.37 (t, 1.9)	7.35 (s, br)	4.8 (s, br) 4.8 (s, br)
**16**	7.24 (s, br)	7.20 (s)	-
**17**	0.86 (d, 6.4)	0.87 (d, 6.6)	0.84 (d, 6.6)
**18a** **18b**	--	4.21 (d,11.3) 3.84 (d, 11.3)	--
**19a** **19b**	4.63 (dd, 2.56, 7.69) 4.33 (d, 7.69)	3.98 (d, 10.6) 3.66 (d, 10.6)	4.68 (dd, 1.9, 7.3) 4.3 (d, 7.9)
**20**	0.89 (s)	0.79 (s)	0.9 (s)

**Table 4 molecules-25-01379-t004:** Anti-inflammatory activity compounds from *B. conferta*.

Compounds	E_max_	ED_50_ (mg/ear)
**1**	71.42	0.3177
**2**	102.04	0.3601
**3**	120.48	0.1286
**5**	149.25	0.3619
**6**	103.09	0.1662

Values are mean ± SEM (n = 5).

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
