# Peer review of "Effect of Terpenoids and Flavonoids Isolated from Baccharis conferta Kunth on TPA-Induced Ear Edema in Mice"

_molecules, 2020, doi:10.3390/molecules25061379_

Round 1
Reviewer 1 Report
Gutiérrez-Román and colleagues wrote an interesting manuscript on the anti-edema effects of some Baccharis conferred phytoextracts. The manuscript has a lot of potential but unfortunately there are some critical issues that should be resolved before publication.
In particular:
- The authors speak of "inflammation" but do not take any tests to evaluate cytokines or other inflammatory mediators. For this reason the authors cannot speak of anti-inflammatory activity. Furthermore, it is not enough to say that the anti-inflammatory action has been evaluated by other authors. Authors should change the title of the manuscript or integrate with molecular data.
- The authors made an excellent phytochemical analysis on the extraction and purification of plant products and a topical treatment that reduces the weight of the tissue damaged by the toxin. In my opinion, reducing the weight of the ear tissue is not enough to establish the therapeutic activity of the substance.
- From an experimental point of view, in my opinion an experimental group is missing in the treatment performed by the authors. In particular, the group of animals that does not receive any treatment is missing. This is important in order to establish how much the weight of the studied tissue fragment varies.
- Can acetone used for topical treatments have harmful effects on the skin of the ear? Has this parameter been measured?
- Have pharmacokinetic and pharmacodynamic studies been carried out?
- I suggest the authors not to use unknown acronyms and numbers in the abstract which should be as clear as possible.
- Finally, I suggest that the authors integrate the introduction by arguing more about the importance of using phytocomplexes and phytoextracts in therapy. In this regard, see Mastinu A et al. Gamma-oryzanol Prevents LPS-induced Brain Inflammation and Cognitive Impairment in Adult Mice. Nutrients; and Premoli M et al. Cannabidiol: Recent advances and new insights for neuropsychiatric disorders treatment. Life Sci. 2019; and Kumar A et al., Cannabimimetic plants: are they new cannabinoidergic modulators? Planta. 2019
Author Response
- The authors speak of "inflammation" but do not take any tests to evaluate cytokines or other inflammatory mediators. For this reason the authors cannot speak of anti-inflammatory activity. Furthermore, it is not enough to say that the anti-inflammatory action has been evaluated by other authors. Authors should change the title of the manuscript or integrate with molecular data.
- The authors made an excellent phytochemical analysis on the extraction and purification of plant products and a topical treatment that reduces the weight of the tissue damaged by the toxin. In my opinion, reducing the weight of the ear tissue is not enough to establish the therapeutic activity of the substance.
- From an experimental point of view, in my opinion an experimental group is missing in the treatment performed by the authors. In particular, the group of animals that does not receive any treatment is missing. This is important in order to establish how much the weight of the studied tissue fragment varies.
- Can acetone used for topical treatments have harmful effects on the skin of the ear? Has this parameter been measured?
- Have pharmacokinetic and pharmacodynamic studies been carried out?
- I suggest the authors not to use unknown acronyms and numbers in the abstract which should be as clear as possible.
- Finally, I suggest that the authors integrate the introduction by arguing more about the importance of using phytocomplexes and phytoextracts in therapy. In this regard, see Mastinu A et al. Gamma-oryzanol Prevents LPS-induced Brain Inflammation and Cognitive Impairment in Adult Mice. Nutrients; and Premoli M et al. Cannabidiol: Recent advances and new insights for neuropsychiatric disorders treatment. Life Sci. 2019; and Kumar A et al., Cannabimimetic plants: are they new cannabinoidergic modulators? Planta. 2019
Authors reply (A word file is attached)
The aim of this work was to find the active compounds from Baccharis conferta, using the TPA induced edema in mice. According with this recommendation, title will be changed by “Effect of Terpenoids and flavonoids isolated from Baccharis conferta Kunth on TPA-induced ear edema in mice”
Authors are agreed with this conclusion. Methods, results, discussion and conclusion of this work are focused on the pharmacological activity through auricular edema in mice induced with TPA.
This observation is correct. However, each treatment includes a reference control. In this edema inhibition model it has been validated and the ears that do not receive treatment are considered as a reference control. The inclusion of experimental animals without any treatment could be repetitive. This work was conducted following international ethic rules that include rational use of animals.
Considering this observation, the phrase “the ears that received acetone did not show any damage or edema” was included in the results section (Histological analysis of ear edema-induced by TPA).
In this work authors describe only pharmacological (pharmacodynamic) effects and chemical characterization of Baccharis conferta. Pharmacokinetic activity will be considered for future perspectives
Acronyms and number were reduced in the abstract section.
This suggestion was considered in the final version of the manuscript.

Reviewer 2 Report
This paper aimed on determining the anti-inflammatory activity of plant baccharis concert Kunth and identifying the compounds responsible for the effect using the TPA-induced ear edema model in mice. It is a topic of interest to the researchers in the related areas and it is the first report of the new compound denominated bacchofertone and compounds 1,2,3,5 demonstrated efficacy at reducing inflammatory parameters as first evidence. The paper only needs few improvement before acceptance for publication. My detailed comments are as follows:
- In abstract part, I suggest to present the detailed data of activity for most significant anti-inflammatory fractions and compounds.
- In page 5, line145, the literature comparison of compound 6 is not clearly described.
- In page 7 and 8, oleanolic acid (4) should be deleted from the title of Figure 4 and 5 because this compound was not tested in this paper. And also, dose response could not be performed clearly from Figure 4.
Author Response
This paper aimed on determining the anti-inflammatory activity of plant baccharis concert Kunth and identifying the compounds responsible for the effect using the TPA-induced ear edema model in mice. It is a topic of interest to the researchers in the related areas and it is the first report of the new compound denominated bacchofertone and compounds 1,2,3,5 demonstrated efficacy at reducing inflammatory parameters as first evidence. The paper only needs few improvement before acceptance for publication. My detailed comments are as follows:
- In abstract part, I suggest to present the detailed data of activity for most significant anti-inflammatory fractions and compounds.
- In page 5, line145, the literature comparison of compound 6 is not clearly described.
- In page 7 and 8, oleanolic acid (4) should be deleted from the title of Figure 4 and 5 because this compound was not tested in this paper. And also, dose response could not be performed clearly from Figure 4.
Authors reply (A word file is attached)
Complete data of the most active treatments were included in the abstract section.
All data were completed in the manuscript.
Oleanolic acid was eliminated from the figures 4 and 5. Correction of legend of figure 4 was done
These data were included in the final version of the manuscript

Reviewer 3 Report
The manuscript “Anti-inflammatory activity of terpenoids and flavonoids isolated from Baccharis conferta Kunth” reports the isolation of 4 clerodane diterpenes, one triterpene and 2 flavones from Baccharis conferta.
The manuscript is well-written, and the work was well-performed (bioassay-guided isolation and structural elucidation). Here I suggest some corrections in order to improve the work before publication.
Abstract section
Line 18: Please replace “clerodano” by clerodane.
Introduction section
Please add more information. There are several studies focused on the ethnopharmacological potential of Baccharis plant species, as well as chemical constituents. A more complete description of the current literature would improve the manuscript. (how many clerodane diterpenes have been described for this plant genus? What is the relationship between this class of compounds and anti-inflammatory activity? And toxicity? Is there any study that points to possible pharmacological sites? Possible mechanisms of action?
Lines 34-38. Please consider replace “One of the pharmacological reports made to this species was the antispasmodic effect of the ethanol extract where they identified the presence of flavonoids and on the other hand the ovicidal effect against Haemonchus contortus from the methanol extract where the active compounds turned out to be isokaemferide and 4,5-di-O-caffeoylquinic”. Here is one suggestion: “Previous report suggested the antispasmodic effect of B. conferta ethanol extract, and the presence of flavonoids. The ovicidal effect against Haemonchus contortus from the methanol extract was also reported, where the active compounds were isokaemferide and 4,5-di-O-caffeoylquinic.”
Results and discussion section
Line 49: Replace “it, promotes” for “,it promotes”. Replace “which promotes the release of…” for “which induces the release of…”
line 59: Please revise “shold”
Line 78: Please consider revise “EI-MS” for “ESI-MS”. This mistake occurs multiple times across the manuscript.
Quasimolecular ion is “a protonated molecule or an ion formed from a molecular ion by loss of a hydrogen atom. The use of the term 'pseudo-molecular ion' is not recommended.”, according to PAC, 1991, 63, 1541. The authors should use only “ion” or “sodium adduct ion”.
Line 94: Remove “On the other hand” (clutter text)
Line 118: Replace “by means of” for “by” (clutter text)
Line 121: In the sentence “…there is no information about any biological effect”, the authors should consider Lee SH et al., PNAS 112, 1733-1738, 2015 DOI: 10.1073/pnas.1424386112
Line 133: Revise “COXY”
Line 145: Consider revise “…isolated cirsimaritin from B. conferta.”
Line 283: Replace “it, was submitted” for “,it was submitted”.
Author Response
The manuscript “Anti-inflammatory activity of terpenoids and flavonoids isolated from Baccharis conferta Kunth” reports the isolation of 4 clerodane diterpenes, one triterpene and 2 flavones from Baccharis conferta.
The manuscript is well-written, and the work was well-performed (bioassay-guided isolation and structural elucidation). Here I suggest some corrections in order to improve the work before publication.
Abstract section
Line 18: Please replace “clerodano” by clerodane.
Introduction section
Please add more information. There are several studies focused on the ethnopharmacological potential of Baccharis plant species, as well as chemical constituents. A more complete description of the current literature would improve the manuscript. (how many clerodane diterpenes have been described for this plant genus? What is the relationship between this class of compounds and anti-inflammatory activity? And toxicity? Is there any study that points to possible pharmacological sites? Possible mechanisms of action?
Lines 34-38. Please consider replace “One of the pharmacological reports made to this species was the antispasmodic effect of the ethanol extract where they identified the presence of flavonoids and on the other hand the ovicidal effect against Haemonchus contortus from the methanol extract where the active compounds turned out to be isokaemferide and 4,5-di-O-caffeoylquinic”. Here is one suggestion: “Previous report suggested the antispasmodic effect of B. conferta ethanol extract, and the presence of flavonoids. The ovicidal effect against Haemonchus contortus from the methanol extract was also reported, where the active compounds were isokaemferide and 4,5-di-O-caffeoylquinic.”
Results and discussion section
Line 49: Replace “it, promotes” for “,it promotes”. Replace “which promotes the release of…” for “which induces the release of…”
line 59: Please revise “shold”
Line 78: Please consider revise “EI-MS” for “ESI-MS”. This mistake occurs multiple times across the manuscript.
Quasimolecular ion is “a protonated molecule or an ion formed from a molecular ion by loss of a hydrogen atom. The use of the term 'pseudo-molecular ion' is not recommended.”, according to PAC, 1991, 63, 1541. The authors should use only “ion” or “sodium adduct ion”.
Line 94: Remove “On the other hand” (clutter text)
Line 118: Replace “by means of” for “by” (clutter text)
Line 121: In the sentence “…there is no information about any biological effect”, the authors should consider Lee SH et al., PNAS 112, 1733-1738, 2015 DOI: 10.1073/pnas.1424386112
Line 133: Revise “COXY”
Line 145: Consider revise “…isolated cirsimaritin from B. conferta.”
Line 283: Replace “it, was submitted” for “,it was submitted”.
Authors reply(A word file is attached)
Abstract section
This change was done
Introduction section
Authors are very grateful with this observation. This suggestion was considered in the manuscript.
These suggestions were considered in the final version of the manuscript.
Results and discussion section
All these typographical mistakes were corrected in the manuscript.

Round 2
Reviewer 1 Report
The manuscript has improved, my doubts and questions have been clarified. In my opinion the manuscript can be published.